# Bacteria and Dry Eye: A Narrative Review

**DOI:** 10.3390/jcm11144019

**Published:** 2022-07-12

**Authors:** Yuchen Wang, Yi Ding, Xiaodan Jiang, Jiarui Yang, Xuemin Li

**Affiliations:** 1Beijing Key Laboratory of Restoration of Damaged Ocular Nerve, Department of Ophthalmology, Peking University Third Hospital, No. 49 North Garden Road, Haidian District, Beijing 100191, China; duo_314@163.com (Y.W.); eyedrjiang@bjmu.edu.cn (X.J.); 2Capital Medical College Attached Beijing Shijitan Hospital, No. 10, Xi Toutiao, Youanmen Wai Street, Beijing 100069, China; dingi0927@163.com

**Keywords:** dry eye, meibomian gland dysfunction, bacteria, microbiome homeostasis, inflammatory factors, mechanisms of dry eye

## Abstract

(1) Background: Dry eye is a multifactorial disease of the ocular surface, the incidence of which has been increasing sharply. The pathogenesis of dry eye, especially in terms of the bacterial flora, has drawn great attention. Additionally, the potential treatment methods need to be explored. (2) Methods: We reviewed more than 100 studies and summarized them briefly in a review. (3) Results: We summarized the bacterial communities found on the ocular surface in the general population and patients with dry eye and found a relationship between dry eye and antibiotic therapy. We identified the possible mechanisms of bacteria in the development of dry eye by discussing factors such as the destruction of the antibacterial barrier, infectious diseases, microbiome homeostasis, inflammatory factors on the ocular surface and vitamin deficiency. (4) Conclusion: We systematically reviewed the recent studies to summarize the bacterial differences between patients with dry eye and the general population and brought up several possible mechanisms and possible treatment targets.

## 1. Introduction

Dry eye is a multifactorial disease of the ocular surface which is characterized by the instability of the tear film accompanied by ocular surface symptoms [1]. At present, dry eye is divided into aqueous-deficient dry eye (ADDE), evaporated dry eye (EDE) and mixed dry eye due to the different pathological processes involved [1]. The incidence of dry eye has been increasing year by year with the progress of science and technology and changes in lifestyle. Eye discomfort caused by dry eye has gradually become one of the most important reasons for clinical visits. The Tear Film and Ocular Surface Society (TFOS) Dry Eye Workshop (DEWS) epidemiology subcommittee conducted a meta-analysis that showed that the overall prevalence ranges from 8.7 to 30.1% for a combination of symptoms and signs [2]. The high prevalence of dry eye is causing large-scale medical expenses, increasing the social burden [3]. Thus, it is of great clinical and social significance to explore the pathogenesis of dry eye and find potential treatment methods.

According to the literature [4], there are many causes of dry eye, such as advanced age, sex, Asian ethnicity, contact lens use, and prolonged exposed to low-humidity environments or air-conditioned rooms; however, some diseases can also trigger dry eye, such as autoimmune diseases. The patients diagnosed with dry eye in Sjögren syndrome were confirmed to have been infected with viruses such as human T-cell lymphotropic virus (HTLV), human immunodeficiency virus (HIV), Epstein-Barr virus (EBV), and hepatitis C virus (HCV). The link between viral infection and dry eye was described by others a decade ago [5]. Rajalakshmy AR et al. [6] also conducted a survey: HCV virus was detected in tears of dry eye patients, and it was found that dry eye was significantly correlated with HCV virus infection.

However, bacterial infections are more common causes of the ocular surface disease. ecently, ocular bacterial flora have drawn great attention in relation to the pathogenesis of dry eye. We know that various infectious diseases, such as anterior blepharitis [7], conjunctivitis [8] and keratitis [9] are related to the occurrence and development of dry eye. Additionally, topical or systemic antibiotics have been proven to be effective in relieving the symptoms and signs of dry eye [10] in patients, while the long-term application of antibiotic eye drops has also been identified as a risk factor for dry eye [11]. All of the above results suggest that alterations in the ocular bacteria may be related to the occurrence of dry eye, while the restoration of the normal commensal flora might be significant in the treatment of dry eye. However, real-world analyses of the correlation between dry eye and ocular bacteria, as well as the molecular mechanisms of bacteria on the ocular surface, remain to be elucidated.

This review summarizes the cultural results of bacteria isolated from the ocular surface in patients with dry eye and provides an opinion on the possible role of bacteria in the pathogenesis of dry eye, aiming to provide a basis for the management of risk factors and the clinical treatment of dry eye, as well as to highlight scientific ideas with potential for future clinical and basic research.

### 1.1. Ocular Surface Flora in General Population

Through the bacterial culture of specimens collected from the conjunctival sac and palpebral margin, recent studies have confirmed the existence of residential bacteria on the ocular surface in the general population, especially preoperative patients undergoing cataract or LASIK treatment.

The bacterial culture of the conjunctival sac has become the routine procedure before intraocular operation in some medical institutions. Suto et al. carried out an investigation on bacterial flora of the conjunctival sac in patients prior to cataract surgery. The bacteria were isolated from 227 (39.2%) of the 579 eyes studied. In total, 191 (67.0%) were Gram-positive cocci, of which coagulase-negative *Staphylococcus* was the most frequent [12]. Additionally, in Hsu et al.’s study, a positive isolation rate was detected in the group of 183 eyes that were cultured, while 85.2% of eyes were positive and gram-negative cocci accounted for the majority of all isolates [13]. The above studies suggest the presence of bacterial flora in the conjunctival sac and verified that the use of antibiotics preoperatively can minimize bacterial growth. In addition, the bacteria flora cultured from the conjunctival sac in the general population have been identified more clearly. In the United Kingdom, Graham et al. detected bacteria in normal subjects, 75% of which were positive in culture, with coagulase-negative *Staphylococcus* being the main isolate [14]. Venugopal et al. also confirmed that the most common isolate was coagulase-negative *Staphylococcus,* with a positive rate of 87.5 % in 16 controls [15]. Similarly, *Staphylococcus* coagulase-negative bacteria (28.6%) were the major organisms in conjunctival swabs in Sierra Leone, and in 276 local residents and 56.8% of swabs in Spain [16,17].

Based on the above studies, with the use of high-throughput sequencing and investigations of the composition and diversity of the bacterial flora, *Staphylococcus*, *Corynebacterium* and *Pseudomonas* have been identified as the principal components on the ocular surface [18]. In most studies, *Staphylococcus* is the most isolated bacterium, changes of which can markedly affect ocular health [19,20,21,22]. In other studies, *Corynebacterium, Pseudomonas* or *Propionibacterium* were the most frequently isolated genera [23,24,25].

The above results show that the positive rates and types of microflora on the ocular surface in the general population varied between studies, which may have been related to regional distribution, ecological environment and ethnic differences, with *Staphylococcus*, *Corynebacterium* and *Propionibacterium* being the common types across various studies, suggesting that these bacteria may play an important role in normal ocular surface function as well.

### 1.2. Ocular Surface Flora in Patients with Dry Eye

According to the locations of bacteria on the ocular surface, the conjunctival sac, palpebral margin and secretion of the Meibomian gland were studied. At the same time, MGD (Meibomain Gland Dysfunction) patients often show signs of anterior blepharitis related to bacterial infection; therefore, some studies were conducted on the ocular surface microflora of MGD patients alone, while most studies did not further classify the types of dry eye [26].

In India, in cultured conjunctival sac samples, 59% were positive, many of which contained coagulase-negative *Staphylococcus* [15]. Meanwhile, the positive rate of bacteria culture in dry eye was 97% in the UK; similarly, the main isolated species were coagulase-negative *Staphylococcus* [14]. A study of the ocular surface microflora of MGD patients in China by Zhang et al. found that the positive rates of bacterial culture in the conjunctival sac was 64.7% for aerobic bacteria and 30.8% for anaerobic bacteria; meanwhile bacteria in the Meibomian gland accounted for 75.6% and 34.3% of aerobic and anaerobic bacteria, respectively [27]. The composition of the bacterial flora presents great diversity in terms of the different populations and areas. Regarding the exploration of the composition of the ocular microbiome in patients with MGD, a number of studies have emerged in the last five years. Jiang et al. (our research team) identified species via RNA sequencing, illustrating that the positive bacterial isolation rate was markedly higher in MG secretions than in the conjunctival sac, while *Corynebacterium macginleyi* was only detected in MGD patients, with an isolation rate of up to 26.3% [21]. It was also found that, when comparing the microfloral composition in MGD patients with that in normal people, *Staphylococcus epidermidis* (aerobes) and *Propionibacterium acnes* (anaerobes) were still the predominant genera [27]. Dong’s study showed an abundance of *Staphylococcus* and *Sphingomonas* in patients with MGD, which were significantly higher than the controls [23]. *Corynebacterium macginleyi* was also observed to have increased isolation rates in other studies, while, after treatment with an intense pulsing light, the isolation rate of *Corynebacterium macginleyi* was significantly inhibited [21,24,27,28].

Additionally, several studies have focused on the bacterial discrepancies in different severity levels of disease conditions. Water et al. in New Zealand reported a positive isolation rate for bacteria in patients with mild MGD of 79.2%, while the number reached 84.8% in patients with moderate to severe MGD [29]. The same phenomenon was observed in a study performed by Jiang et al. (our research team), with a positive isolation rate of 89.5% in the severe MGD group and rates of 62.5% and 58.2% in mild and moderate MGD groups, which were deemed statistically significant differences. To date, most of the studies have mainly been descriptive studies. The correlations between bacterial characteristics and the severity of dry eye are far from sufficient. A new grading system for the bacterial severity of MGD has been proposed by our team, whereby the bacterial composition can be divided into four grades, with grade 1 meaning no positive culture results, grade 2 meaning only *S. epidermidis* was isolated, grade 3 representing positive results for both *S. epidermidis* and other bacteria, and grade 4 indicating only other bacteria. This grading system was established to explore the relationship between the bacteria and the severity of MGD, which appeared to be positively correlated with a higher grade of bacterial severity. The microbiomes of the general population and dry eye patients over the last five years are shown in Table 1.

We can conclude that the most commonly isolated bacteria on the ocular surface of MGD patients are similar to the general population, namely coagulase-negative *Staphylococcus*; however, the population of dry eye patients showed higher bacterial diversity and positive isolation rates, indicating that the normal flora on the ocular surface may also be involved in the disease progression of patients with dry eye, while the normal flora can be significantly influenced by pathogens with the progression of dry eye.

In view of the above, to study the qualitative and quantitative relationships between the bacteria and dry eye, more studies are needed to further explore the possible mechanisms of such differences, along with a standard evaluation of the severity of dry eye disease.

### 1.3. Dry eye and Antibiotic Therapy

Several clinical studies have been conducted to explore the clinical efficacy of antimicrobial agents in patients with dry eye, and treatment of MGD with ocular surface antibiotics has become the clinical guideline [30,31]. Topical therapy with azithromycin on the ocular surface and oral therapy with azithromycin have been proven to be effective in relieving symptoms [30,31]. The possible mechanisms of azithromycin in treating MGD may include various aspects. First of all, azithromycin is a broad-spectrum macrolide antibiotic, which has favorable tissue penetration and possibly potent ocular anti-inflammatory properties. Additionally, azithromycin could suppress bacterial lipases, which are thought to degrade the lipids in the Meibomian gland, so as to upgrade lipid secretions to treat dry eye induced by MGD. On the contrary, if antibiotics are not used, bacteria lipases will degrade the lipids, which can lead to abnormal secretions of the Meibomian gland. Furthermore, an abnormal meibum causes a higher melting point and lipid sclerosis, which is clinically characterized by ductal plugging and secretion dysfunction [32]. Secondly, azithromycin suppresses conjunctival inflammation, which may block the activation of nuclear factor-kappa B (NF-κB), leading to decreased inflammatory cytokine levels such as interleukin-6 and -8 to improve dry eye symptoms [33]. Finally, azithromycin can act directly to stimulate the accumulation of cholesterol, phospholipid and lysosomes in Meibomian gland epithelial cells. This process is a common adverse effect in other conditions, but has a potential beneficial effect on MGD [34]. Additionally, a Meibomian gland massage in combination with levofloxacin also had significant antibacterial effects, which reduced the bacteria growth from 38.5% to 19.8% in the conjunctival sac, and from 38.5% to 11.0% in Meibomian glands [27]. The above results suggest that topical levofloxacin has antibacterial effects, which could be a potential treatment option for patients with MGD [20].

Moreover, other studies have compared the efficacy of different types of antibiotics. Kashkouli et al. conducted a randomized double-blind clinical trial to compare the efficacy of oral azithromycin for 5 days and oral doxycycline for 1 month, with the results showing that the symptoms and signs improved significantly in both groups of patients, while the side effects and clinical adverse events of azithromycin were lower than those of doxycycline [35].

Except for antibiotics, a number of foods and drugs with antibacterial effects have been shown to significantly improve symptoms and signs in patients with dry eye. For example, the topical application of antibacterial honey on the ocular surface has been proven to improve the symptoms and signs, and it has also been proven to reduce the symptoms of dry eye in contact lens wearers [36,37]. In addition, chitosan, having antibacterial effects, has also been proposed for use in artificial tears to treat dry eye [38].

The above findings suggest that antibacterial drugs can effectively improve the signs and symptoms of dry eye patients, especially for the MGD. These results also confirm the correlation between dry eye and bacterial infection, although further studies still need to be performed to explore the correlations between the symptoms, signs and different bacteria before and after treatment.

### 1.4. Possible Mechanisms of Bacteria in the Development of Dry Rye

Based on the current research, we concluded that dry eye is associated with the changes in bacterial flora on the ocular surface. In this section, we explore the possible mechanism of bacterial changes involved in the dry eye progression.

The destruction of the antibacterial barrier, contact lens wear, infectious diseases and treatment side effects are discussed in the context of clinical pathogenesis. We also focus on the cellular and molecular pathways, including disturbances of microbiome homeostasis, toll-like receptor (TLR) activation and its downstream inflammatory factors, and vitamin deficiencies. The relationship between bacteria and dry eye and the possible effects caused by the above factors are shown in Figure 1.

#### 1.4.1. Destruction of Antibacterial Barrier on the Ocular Surface

The normal ocular surface in humans is a complete microbial defense system based on anatomical and physical functions, which mainly includes three parts: physiological defense, biochemical defense and immune defense.

Physiological defenses prevent microbial invasion through the human anatomical structure, physiological reflection and metabolism, and there are several physiological defenses on the ocular surface. One of them is the eyelashes, which can filter dust and pollution, while the blink reflex can keep foreign bodies or insects away from the ocular surface. Moreover, blinking means microbes are able to be absorbed at the lacrimal point from the ocular surface and further discharged from the lacrimal duct and nasal cavity. Meanwhile, epithelial cells on the ocular surface also play an important role. The epithelial cells form a barrier that prevents microbial invasion into the cornea and conjunctiva. Additionally, exfoliation of the squamous epithelium also reduces microbial populations [36]. These physiological defense barriers can prevent the invasion of microorganisms and protect the structure and function of the ocular surface.

The biochemical defense process is closely related to immune defense, mainly involving biological barriers composed of antibacterial substances in tears and on the ocular surface. These antibacterial substances mediate the downstream immune response to form an immune barrier through the recognition of bacterial non-specific antigens. Previous studies have found antibacterial ingredients in tears, such as lysozyme, lactoferrin, tear lipocalin, secretory immunoglobulin A and complement C3. Lysozyme and lactoferrin, which are secreted by the main lacrimal and accessory lacrimal glands, account for 20–30% of the protein in basal and reflex tears. Lysozyme can kill Gram-positive bacteria by hydrolyzing cell walls [39]. Lactoferrin, which has a high capacity to bind divalent cations (e.g., iron), can deprive bacteria of the essential nutrients needed for growth. Furthermore, a highly basic sequence at the N-terminus of lactoferrin can act as a cationic detergent to disrupt the cell membrane. It is commonly believed that lactoferrin plays a key role in maintaining ocular stability [39,40]. Tear lipocalin, which is produced by the acinar cells of the main lacrimal gland, represents nearly 25% of the protein component in reflex tears [41]. Some isoforms of lipocalin contain a protease inhibitory domain and protect the ocular surface from cysteine proteases of bacteria [39].

The major antibody in the tear film is secretory immunoglobulin A (sIgA), which is produced by plasma cells of the main and accessory lacrimal glands associated with lymphoid tissues. Furthermore, sIgA not only prevents the bacterial attachment to host cells via specific antigen binding sites, but also binds to lectin-like adhesin molecules to aggregate and subsequently remove pathogens. Additionally, sIgA has also been proven to induce the aggregation of phagocytic neutrophils to further kill bacteria [39]. Complement C3 is another main component of antibacterial ingredients, which mainly comes from the exudate of conjunctival vessels and the synthesis of conjunctival corneal epithelial cells. Activation of the complement pathway can induce acute inflammation responses, such as fragments from complement pathways, as opsonin facilitates neutrophils to recognize targets; this results in the formation of membrane attack complexes. Additionally, recent studies have reported that Meibomian lipids possess Gram-positive and Gram-negative bacterial abilities, while mucins produced by goblet cells can maintain homeostasis and regulate immune tolerance. The above results indicate that the immune response networks constitute the antibacterial barrier and play an important role in the defense of microbial invasion on the ocular surface.

Under a series of risk factors, the microbial defense system is damaged, which may further lead to the proliferation of microorganisms on the ocular surface, resulting in alterations of the ocular surface flora.

As lifestyle changes, electronic devices and mobile terminals play important roles in modern life, video display terminal (VDT) syndrome is a leading cause of dry eye. More time spent in front of screens leads to less blinking and lower blink reflex ability, meaning foreign bodies and microorganisms are not discharged effectively, and further enrichment and proliferation of microorganisms occurs. In addition, corneal spot staining is a common sign in patients with dry eye, which may be related to having a shorter tear break-up time (TBUT) and the failure to form an effective tear film on the ocular surface. Previous studies using corneal confocal microscopy also found that some patients did not have corneal staining, but that the density of corneal epithelial cells decreased [42,43]. Patients with dry eye always display corneal epithelium defects, which indicates that the corneal defense barrier is damaged and that its antibacterial effects may be decreased accordingly.

Regarding the biochemistry and immune defense, on the one hand, previous studies have shown that the lacrimal functional unit (LFU) plays a crucial role in tear secretion and quickly responds to the environment to activate the nerve reflex, allowing the regulation of tear secretion [44]. In the early stages of dry eye, in both ADDE and EDE, tears secreted by the lacrimal gland can be stimulated by the LFU in a compensatory way. However, with the development of dry eye, the neural reflex is abated and tear secretion is reduced. In turn, dry eye is more serious, while, at the same time, antimicrobial substances secreted by the lacrimal gland may also be reduced, which can lead to a decreased ocular surface and non-specific immune ability. On the other hand, as the severity of MGD increases, the obstruction of the Meibomian gland orifices is aggravated, which alters the composition and capacity of Meibomian gland secretions, meaning that the lipid layer cannot be evenly distributed on the ocular surface and that the antibacterial effects are insufficient.

When the concentration of non-specific immune antibacterial substances decreases, the proliferation of pathogens will activate the downstream inflammation pathway. This will be discussed in detail in terms of toll-like receptor activation, but it is worth noting that goblet cells play an important role in immune regulation, as the density of conjunctival goblet cells and their related function is affected in patients with dry eye [45,46]. This pathological process may enhance the immune response, causing damage to the ocular surface and further exacerbating symptoms step-by-step through stronger downstream inflammation.

#### 1.4.2. Effects of Infectious Diseases and Available Treatments

Moreover, in addition to the distribution of bacteria on the ocular surface in dry eye patients, there is a wide range of patients with dry eye who often show symptoms after the treatment of bacterial infection of the ocular surface, among which conjunctivitis is the most common. Huang et al. examined 73 eyes with acute onset of conjunctival injection and found that the tear film stability decreased obviously after infection and changed significantly after recovery from acute conjunctivitis, when abnormal break up time (BUT) measurements, a Schirmer I test (SIT), a tear meniscus height (TMH) test and fluorescein staining (FL) were performed in the early period; however, at 30 days after recovery, the variables returned to normal [8]. Moreover, other studies have shown normal tear volumes in patients with chronic conjunctivitis, while BUT values were significantly decreased [47]. The above results suggest that antibiotic treatment and the disease state of conjunctivitis may promote the occurrence and development of dry eye, the possible mechanisms of which are as follows. First of all, pathogens may directly attack the conjunctival and corneal epithelial cells, damaging the epithelial cells by producing toxic substances and inflammatory mediators in the acute or chronic stages of conjunctivitis. Secondly, the conjunctival epithelium and goblet cells are closely related to the secretion of tears and mucin, meaning that the damage to epithelial cells may directly lead to abnormal tear components and may promote the occurrence of dry eye. Furthermore, frequent use of anti-inflammatory or antibiotic eye drops could flush out and dilute the lipids and proteins in the tear film, resulting in an abnormal tear composition. The preservatives in eye drops will also result in corneal and conjunctival epithelial injury due to their toxic effects. In summary, dry eye after conjunctivitis treatment may be related to both the disease state of conjunctivitis and the application of drugs in the treatment process. It is necessary to conduct regular follow-ups for these patients in clinical practice.

#### 1.4.3. Contact Lens Wear

Corneal contact lenses, which are used for refractive correction and have become a common clinical therapy, are an important cause of damage to the corneal epithelial barrier [48]. Dogan et al. compared 29 long-term users of soft corneal contact lens and 14 normal controls and found that the symptoms and signs of dry eye in the experimental group were more serious than those in the control group [49]. Chen et al. obtained similar results in their study, where, compared with those who do not wear corneal contact lens, patients wearing contact lens showed higher rates of dry eye symptoms [50]. In addition, different types of corneal contact lenses have different effects on dry eye; mild to moderate dry eye was more frequently observed in soft corneal contact lens wearers than those wearing rigid corneal contact lens [50]. Meanwhile, Shi et al. conducted a study that showed that the osmotic pressure in tears increased significantly after 3 and 6 months of wearing contact lenses; however, no significant difference between soft corneal contact lens and semi-rigid corneal contact lens wearers was found [51]. The above results suggest that wearing corneal contact lenses may affect the stability of the tear film and further induce the occurrence of dry eye.

The enhancement of tear evaporation is the reason for dry eye caused by contact lens wearing, the possible mechanism of which includes various aspects. First of all, the wearing of corneal contact lenses will affect the spread of the tear film, separating the tear film into the pre- and post-lens tear film. The lipids in the pre-lens tear film are easily deposited on the front surface of the contact lenses. Secondly, contact lenses move along the conjunctiva with every blink, which affects the stability of the tear film. Thirdly, the incomplete blinking caused by wearing contact lens is increased, meaning that the tear film is not distributed effectively and evenly, causing decreased stability of the tear film. Finally, different types of contact lenses have different mechanisms of dry eye development, with the higher water content of soft contact lenses and the wider range of mobility of rigid contact lenses resulting in worse tear stability and stronger tear evaporation effects [51,52,53].

There are also relevant reports on the relationship between contact lens use and flora changes on the ocular surface. Some studies found that the positive rates were enhanced with contact lens use. For example, Larkin et al. found that the positive isolation rate of bacterial culture after wearing corneal contact lenses reached 51.8% [54]. The same results were also obtained by Liaqat et al., who showed that the positive rate increased to 79%, while *Bacillus* was the most common bacteria with a positive rate of 26% [55]. Additionally, hemolysin production, serum resistance and hemagglutination are the most common pathogeneses, which make up 65%, 61% and 56% of the pathogenic characteristics of the bacteria. Multiple antibiotic resistance was also thought to be common, and was recorded in 66.13% of isolates. According to Boost et al., no significant differences were found between the conjunctival microorganisms of lens and non-lens wearers, but lens wearers had more similarities between the conjunctival and lower lid microbiota than non-lens wearers; thus, the ocular environment may be changed and disturbed as a consequence [41]. In addition, the corneal infiltrative events (CIES) were explored in lens wearers, with the results showing that 89% of the contact lenses cultured had a bacterial count of ≥10^4^–10^8^/mL. *Achromobacter* spp. was most frequently identified. Further bacteriostatic experiments on *Achromobacter* spp. isolated from patients with CIES and cultured in disinfecting solution for 14 days still showed antibiotic resistance [56]. Additionally, severe ocular surface diseases such as bacterial keratitis were also significantly associated with wearing contact lenses. Obrubov et al. revealed that 79.3% patients in 77 cases of severe bacterial keratitis had a history of wearing contact lenses, with statistics showing that wearing bandage contact lenses is an independent risk factor for keratitis [57]. The above results suggest that wearing contact lenses has an important correlation with flora changes of the ocular surface.

Nowadays, next-generation sequencing is becoming more and more widely used, according to Okonkwo A et al. [58] *Acinetobacter*, *Corynebacterium*, *Propionibacterium*, *Staphylococcus* and *Streptococcus* were detected in most of the studies, and contact lens wearers had significantly higher levels of *Pseudomonas, Acinetobacter, Methylobacterium* and *Lactobacillus*.

Regarding animal research, Matteo et al. built a CD11c-YEP mice model. In those wearing contact lenses for 24 h, corneal CD11c-positive cells were greatly changed in both distribution and density. Although no direct inoculation of bacteria on contact lenses was applied before wearing them, bacteria on the surfaces of contact lenses were still observed 24 h after wearing, mainly comprised of the natural flora of the conjunctiva and skin. A subgroup of mice wore contact lenses for at least 5 days, and the anterior corneal surface showed vacuole-like changes. The migration of macrophage cells was detected in the corneal stromal layer resulted in morphological changes of corneal stromal cells [59].

Regarding the mechanisms related to the bacterial changes following changes to the ocular surface, several assumptions are raised in this section. Wearing contact lenses decreases the blinking function, which disturbs the tear circulation and leads to the stagnation of tears under the contact lenses. As such, the oxygen and nutrients provided by the tears are less concentrated and the growth of bacteria changes due to the corresponding aspects. On one hand, the decreased oxygen and nutrient concentrations in the tear lead to changes in the microenvironment on the ocular surface, while increased anaerobic metabolites enhance the toxicity and damage to corneal and conjunctival epithelial cells, resulting in the destruction of the ocular surface barrier, which allows bacteria to proliferate. The metabolism of corneal and conjunctival epithelial cells under such anoxic circumstances, on the other hand, can cause the accumulation of carbon dioxide and local pH reductions, leading to epithelial edema and injury, which further impairs the tight connections between epithelial cells and promotes the invasion and proliferation of bacteria. In addition, contact lenses as foreign bodies on the ocular surface can become a cultural medium for bacteria and can provide a location for adhesion, leading to the proliferation of bacteria on the ocular surface. *Pseudomonas aeruginosa* keratitis is the most frequently reported complication from wearing contact lenses. Recent studies have revealed that the hydrophobicity and adhesive ability of *Pseudomonas aeruginosa* are significantly higher than in other species, which might explain the high pathogenicity in contact lens wearers [56]. *Pseudomonas aeruginosa* adheres to sediments on the lenses, while TLR4 and TLR5 on macrophages recognize the flagellin and lipopolysaccharides in *Pseudomonas aeruginosa* and initiate the myeloid differentiation primary response gene 88 (MyD88)-mediated pro-inflammatory pathway. The above pathophysiological studies suggest that contact lens wearing may cause changes in ocular surface flora by affecting the environment, while further studies are still needed to clarify the mechanism.

The above explains the relationship between wearing contact lenses and bacterial infection in terms of the mechanism, as wearing contact lenses can destroy the barrier of the ocular surface, inducing the infiltration of inflammatory cells and bacterial aggregation. It is necessary to further clarify the molecular pathway involved in wearing contact lenses and to design relevant therapeutic drugs to guide clinical practice.

#### 1.4.4. Microbiome Homeostasis

According to the above studies, bacteria can trigger local immune and inflammatory reactions on the ocular surface. We wondered whether there is a balance between the quantity and variety of the bacteria on the ocular surface, and whether bacteria-related dry eye can be attributed to breaking this balance. Our research team first suggested a bacterial severity grading system, and our results indicated that, as the severity of MGD increased, the isolation rate of the commensal bacteria, *Staphylococcus epidermidis*, was significantly decreased, which proved that the progression of dry eye greatly disturbed the bacterial balance. However, the role of commensal bacteria in the normal ocular surface remains unclear. The animal research conducted by Leger et al. provided some insight. The study identified that *Corynebacterium mastitidis* was a commensal organism found in both humans and mice, which was able to uniquely colonize on the ocular surface. Under certain physical conditions, colonization of *Corynebacterium mastitidis* promoted the production of IL-17 by γδ T cells in the ocular mucosa, conferring protection of the cornea from *Candida albicans* and *Pseudomonas aeruginosa* invasion. When applying gentamicin gel topically, which is susceptible to *Corynebacterium mastitidis* in vitro, significant reductions in γδ T cells and IL-17 were observed, as well as in its downstream effectors Cxcl1, -3 and -10. A decrease in neutrophil recruitment to the ocular mucosa was also observed [60].

The topical use of general bacteria on the ocular surface was also brought up in other studies. To treat patients with irritable eye syndrome (IES)-related ocular surface disorder, *Lactobacillus acidophilus* lysates were used topically and proved effective in decreasing the lipopolysaccharide (LPS)-stimulated macrophages by inhibiting IL-1β and TNF-α in a dose- and size-dependent manner. The possible mechanism behind this phenomenon might be explained by the suppressive effects involved in blocking the pathogenesis induced by other bacteria [61]. This study suggests that commensal bacteria could induce and regulate the corresponding immune response, inhibiting the colonization of foreign pathogens.

The studies above have shown that resident bacteria and lysates can inhibit the colonization of other bacteria on the ocular surface and reduce inflammation. We speculated that with the development of dry eye, the microenvironment of the ocular surface changed, leading to the inhibition of resident bacterial growth, although its inhibitory effect on other pathogens decreased at the same time and those pathogens began to colonize on the ocular surface, resulting in a higher concentration of harmful metabolic products, which in turn aggravated the dry eye severity. This hypothesis is also consistent with the above clinical findings.

In the aggregate, those results suggest that resident ocular commensals can maintain ocular immune homeostasis, and that the disturbance of the microbial balance has consequences for the ocular disease [62]; however, experiments are still needed to demonstrate our hypothesis in the development of dry eye.

#### 1.4.5. Ocular Surface Toll-like Receptor (TLR) Activation and Inflammatory Factors

The ocular surface is an exposed mucosa. It has many immune cells such as natural killer cells, dendritic cells and macrophages, as well as CD4 and CD8^+^ T cells that are activated when the barrier is invaded by microorganisms [63]. Some minor alterations stimulate metalloproteinase production, inflammatory cell recruitment, dendritic cell maturation, and activation of an adaptive T-cell mediated response.

TLRs are membrane-binding proteins that are known to recognize conserved motifs on pathogen-associated molecular patterns (PAMPs) on bacteria, viruses, fungi and protozoa to further upregulate the expression of inflammatory factors and chemokines. When ocular surface inflammatory substances change and are detected by TLR, TLR is activated, resulting in a T-cell-mediated downstream inflammatory response that releases inflammatory cytokines.

TLRs are expressed on a variety of cells, including the epithelia, endothelia, antigen-presenting cells and lymphocytes in conjunctival and corneal tissues. TLR2, -3, -4, -5 and -7 are widely found in the conjunctiva, corneal sclera margin and corneal epithelial cells [64,65]. In general, TLR2/1, TLR2/6, TLR4 and TLR5 are present on the cell membranes, whereas TLR3, TLR7, TLR8 and TLR9 are localized on endosomes in ocular cells [65,66]. Generally, TLRs can protect the ocular surface from microbial infection. After the activation of TLRs by pathogens on the ocular surface, cytokines and chemokines are synthesized, stimulating the aggregation of immune and inflammatory cells, which alleviates the microbial load. Meanwhile, TLR may facilitate the activation of acquired immune response by enhancing MHC II and co-stimulatory molecule expression on antigen-presenting cells resident in the cornea and conjunctiva [67,68].

It has been demonstrated that different members of the TLR family have different roles. TLR2, TLR4, TLR5 and TLR9 have been shown to play key roles in *P. aeruginosa-* and *S. aureus*-induced corneal inflammation [66]. Studies using a mice model of *S. aureus*-induced corneal inflammation revealed that the immune response of TLR2- and MyD88-deficient mice was significantly reduced compared with that of TLR4- and TLR9-deficient mice, suggesting that saLP (*S. aureus* bacterial lipoproteins) activates TLR2 and MyD88 to trigger the innate immune response and TLR4 responses via the production of proinflammatory cytokines and defense molecules [69,70]. Additionally, the protein and mRNA levels of TLR9 were increased in conjunctival, limbal and corneal epithelial cells. C57BL/6 mice were more easily infected by *Pseudomonas aeruginosa* and the C57BL/6 model further underwent corneal perforation. The possible mechanism is that TLR9 mediates the immune response of Th1 cells, with the ability to remove bacteria, which also leads to corneal injury. As a result, TLR9 mRNA was downregulated during bacterial infection. Meanwhile, TLR4-deficient mice also presented susceptibility to *Pseudomonas aeruginosa*. The expression of TLR4 on the ocular surface during disease was enhanced, showing that TLR4 plays a certain role in the suppression of *Pseudomonas aeruginosa* [71,72]. TLR5 could only recognize flagellin from Gram-negative bacteria localized to basal and wing-cell layers of the corneal epithelium.

According to the above, it was hypothesized that not only do different bacteria utilize specific mechanisms to initiate the host’s innate immunity, but also the diversity and expression levels of TLRs at the ocular surface in individuals have various effects on the intensity of the inflammatory response caused by TLR stimulation [66].

Moreover, to investigate the alterations in TLR expression in dry eye, Racbel et al. conducted an experiment and found that TLR2, TLR3, TLR4 and TLR9 mRNA expression levels were upregulated in the palpebral conjunctiva in experimental dry eye models. Additionally, upregulated levels of TLR2, -3 and -9 in the corneal epithelium and TLR2 and -5 in the lacrimal gland were also observed [73]. Similar results were also obtained by Hyun et al., whereby TLR4 protein was significantly increased in the corneal stromal cells but not in epithelial cells [74]. Intravenous injection of TLR4 inhibitor in a mouse model decreased the severity of corneal fluorescein staining and reduced the mRNA levels of downstream components such as IL-1β, IL-6 and TNF, as well as the infiltration of CD11II [74]. In a mouse model of ADDE [75], MyD88-deficient mice showed less fluorescein staining along with significant decreases in MMP-2, -3 and -8, which are regulated in the TLR pathways. In human experiments, the studies on TLR are more extensive with Sjogren’s syndrome. Using immunohistochemical staining methods, TLR2, TLR3, TLR4 and MyD88 were found to be increasingly expressed in the salivary cells of patients with Sjogren’s syndrome (SS) [76,77]. The same results can be observed in tissues obtained by salivary biopsy in SS, in which TLR1, -2, -3 and -4 were increased, while CD54, CD40, MHC I, TLR8 and TLR9 were also upregulated [78,79,80]. Until now, differences in TLR expression have not been found in ocular surface specimens, which may be due to the limitations of in vivo sampling. Moreover, Rachel et al. cultured a SV40 human corneal epithelial cell line and a normal human conjunctival epithelial cell line under hyperosmolar stress, discovering that TLR4 and TLR5 mRNA were significantly upregulated while TLR9 mRNA was downregulated, which indicated that hyperosmolar stress influences TLR expression on the ocular surface [81].

These studies revealed that inhibition of the TLR pathway could reduce the ocular surface damage in dry eye progression. In summary, there are significant changes in the transcription and expression levels of TLRs in the cells in dry eye, which may be related to the corresponding changes in the environment of the ocular surface and may further affect the ability for bacterial recognition and antigen presenting.

After the combination with TLRs, bacterial antigens are recognized, leading to the activation of the NF-κB pathway, which has been widely accepted as being important in the production of downstream effector molecules [64]. NF-κB contains a wide variety of downstream effectors, which are mainly inflammatory factors, such as MCP-1, IL-8, TNF-α, IL-6, IL-10 and IL-11, as well as MMP-1 and MMP-3 [82]. Liposaccharides (LPS) can be recognized by TLR4 to further activate NF-κB and stimulate the production of IL-6 and IL-8. S. aureus protein A induces the inflammatory response of NF-κB through the activation of TLR2, and expresses IL-8 and TNF-α [82]. Moreover, the presence of the NF-κB inhibitor blocks P. aeruginosa-induced NF-κB activation and inhibits the expression and secretion of IL-6, IL-8 and TNF-α [83]. Meanwhile, the correlation between the severity of dry eye and the level of inflammatory factors has been confirmed by a large number of studies, such as the work performed by Lam et al., who compared the concentrations of tear cytokines in ADDE and control patients, with the results indicating that IL-6, IL-8 and TNF-α were significantly higher, while in patients with MGD, IL-12 was significantly higher and IL-6 was positively correlated with irritation symptoms [84].

Additionally, Th17 differentiation and the upregulation of IL-6 and IL-23 in the draining lymphoid tissue were induced by hyperosmolar stress, as reported in Fan et al.’s research [85]. Following the differentiation and expansion, Th17 cells were activated by antigen-presenting cells (APCs) and by being exposed to certain cytokines, specifically IL-1β, IL-6, IL-12 and TGF-β. The APCs then migrated to the ocular surface by forming a complex with CCR6^+^Th17 and facilitated by the attractant chemokine CCL-20 at the ocular surface, resulting in elevated levels of MMPs and IL-17A at the ocular surface, which are related to corneal epitheliopathy and the disruption of the barrier function. Moreover, the cytokines mentioned above also contribute to lymphangiogenesis and B-cell proliferation, which further increase ocular surface inflammation [85]. Overall, Th17 and IL-17A play important roles in the downstream pathway of NF-κB and mediate the immune response in chronic inflammatory ocular disease.

Interferon, a class of cytokine produced in response to invasion by viruses, pathogens and neoplastic cells, plays an irreplaceable role in innate and adaptive immunity. Preliminary studies [86] have found that IFNs (Interferons) play an indispensable role in the occurrence of dry eye. Zhang et al. [87] immunostained IFN-γR to evaluate the expression of IFN-γR in the conjunctiva and found that IFN-γR was expressed in all cell layers of the conjunctival epithelia in a dry eye mouse model. Pflugfelder et al. [86] noted that IFN-γ decreases the conjunctival goblet cell density in mice. A similar result was also obtained by Ogawa et al. [88], who detected the levels of interferon in a mouse model of dry eye and found a significant increase. Moreover, in human studies, the levels of IFN-γ tested through tear samples significantly increased in dry eye patients according to Massingale et al. [84], while the expression of IFN-γ receptors also increased in patients with ADDE [86].

After the activation of natural and innate immunity by pathogens such as bacteria and viruses, IFNs are involved in the entire process of immunity. IFN-α and IFN-γ are primarily secreted from natural killer cells (NK cells) and act to enhance inflammation. In turn, they stimulate the production of DC cells, NK cells and T and B cells via B-cell activating factor. Then, after identifying the MHC on the surfaces of the DC and NK cells, T cells secrete IFN-γ and IFN-α and stimulate the production of toxins through the NF-κB pathway to cause damage to the lacrimal gland cells and ocular surfaces. Additionally, IFN-γ has been reported to aggravate goblet cell loss through the suppression of IL-13 signaling via multiple mechanisms, such as interrupting the expression of the SAM-pointed domain epithelial-specific transcription factor (SPDEF), which is essential for goblet cell differentiation, epithelial apoptosis and keratinization of the conjunctival epithelium via the JAK1/2 pathways, as shown in dry eye models. Moreover, IFN-γ is an important contributor to squamous metaplasia of goblet cells. Eventually, these factors work together and lead to dry eye disease [88].

In summary, studies have shown that the activation of TLR and IFN is associated with the expression of the NF-κB pathway and its downstream inflammatory cytokines, while ocular surface bacteria and their metabolites may upregulate inflammatory cytokines through this pathway, further causing the symptoms of dry eye (Figure 2).

#### 1.4.6. Vitamin Deficiency

Vitamins are multifunctional hormones, which have irreplaceable effects in terms of maintaining the normal function of the metabolism and body health. Recent studies have revealed that vitamins have great impact in the prevention and treatment of ocular surface diseases. The possible mechanisms of the vitamins involved in bacterial proliferation, metabolism and the relationships with dry eye progression are discussed in this section.

##### Vitamin B Deficiency

Vitamin B is a water-soluble substance that is present in food, especially in meat, dairy products and animal livers. Vitamin B deficiency was found to be associated with various ocular diseases, including neuropathic ocular diseases and age-related cataracts, suggesting that vitamin B plays an important role in ocular health. There is evidence that a deficiency of riboflavin, also known as vitamin B2, can decrease the microvilli and microplicae in the superficial epithelium of the conjunctiva, leading to a decreased number of goblet cells [89]. According to Yang et al.’s research, in patients with DED, VB12 nebulization appeared to be effective in improving the symptoms and signs of dry eye, although with elevated BUT levels [90]. Moreover, VB12 may contribute to the relief of oxidative stress. Macri et al. investigated three cohorts of patients who underwent planned cataract surgery. Patients with dry eye receiving vitamin B12 eye drops before the surgery presented significantly reduced levels of oxidative stress and OSDI, and increased Schirmer’s test scores and BUT levels compared with patients without dry eye and patients with dry eye receiving no treatment [91].

We also noticed that α-lipoic acid (ALA), a member of the vitamin B family, is a strong antioxidant that could reduce NFAT5, which is the nuclear factor of the activated T cells–NF-kB axis, in order to induce the inflammation of lacrimal glands and ameliorate dry eye symptoms [92]. Moreover, experimental studies [93] proved that ALA can regulate the expression of MMP-9 to degrade the corneal epithelial cells, increase the superoxide dismutase (SOD), catalase (CAT) and glutathione peroxidase (GPx) levels to prevent the MMP activation, and enhance the antioxidant defense by activating the Nuclear factor erythroid-2-related factor (Nfr-2).

All in all, ALA is capable of preventing dry eye by alleviating the inflammation in the ocular surface. The studies mentioned above demonstrate that vitamin B deficiency is associated with the occurrence of dry eye, while the direct relationship between bacteria and ALA on the ocular surface is still unknown.

##### Vitamin D Deficiency

Vitamin D is a lipid-soluble vitamin that has been well studied during the past decade. Health problems are increasingly thought to be linked to vitamin D deficiency. Eye disorders are no exception, including optic neuritis and myopia. Vitamin D appears in the tear film, and it is thought to be essential for the tear film’s stability. According to Pelin et al., in women with VD deficiency, 52–74% had dry eye. Vitamin D levels were negatively correlated with OSDI and positively correlated with Schirmer’s test and TBUT scores [94]. More studies are showing that serum vitamin D and calcitriol concentrations are positively correlated with tear secretion and TBUT, while intramuscular injection of vitamin D is effective for the treatment of dry eye. Studies have shown that the efficacy of drugs for treating dry eye, such as lipid-containing artificial tears (CLAT) and hyaluronate (HU), depends on the 25-HD levels. Hwang et al. divided patients with dry eye into a VD deficiency group and non-VD deficiency group. Both groups were provided with CLAT and HU, and it was found that the TBUT, corneal fluorescein staining and Schirmer test scores remained unchanged in the VD deficiency group, whereas those in the non-VD deficiency group were improved. As a result, the effects of topical CLAT and HU were dependent on serum 25-HD levels, as cholecalciferol enhanced the efficacy of topical treatment [95].

Moreover, vitamin D has potential for the management of inflammation in dry eye. Vitamin D has been reported to repress the responses of both Th1- and Th2-type cells to reduce ocular surface inflammation. An increase in dendritic cell density (DCD) was observed in evaporative dry eyes, and an inverse correlation was observed between vitamin D and DCD with dendritic processes [96]. Calcitriol [97], known as the 1,25-dihtdroxy-vitamin D3, is well known to inhibit inflammation, mainly through its suppressive effects on lymphocyte proliferation, cytokine and chemokine expression, and antigen-presenting cell differentiation.

In in vitro studies, calcitriol could inhibit *Pseudomonas aeruginosa* infection and activate the corneal epithelial cell Toll-like receptors [98]. Meanwhile, topical use of calcitriol in mouse models of *Pseudomonas aeruginosa* infection effectively inhibited the secretion of inflammatory cytokines, such as TNF-α and IL-1 [97]. Yin et al. [97] reported that the corneal endothelial cells contained vitamin D receptors, while vitamin D metabolites were found in both aqueous humor and vitreous bodies. Calcitriol enhanced the function of the corneal endothelial barrier by upregulating the tight junction protein and zo-1, while the corneal thickness of the mice with vitamin D receptor knockout was decreased. Lu et al. demonstrated that tear fluid produced by lacrimal and para-lacrimal glands contains megalin and cubilin molecules that transported VD. Zhang et al. [97] treated dry eye model mice with a solution of calcitriol. The corneal fluorescein staining, tear volume, inflammation index and TBUT scores changed obviously; the tear volume was significantly higher than in the control group and the corneal epithelium was smoother in the calcitriol treatment group, meaning calcitriol can improve the stability of epithelial and tear films.

We hypothesized that the topical application of vitamin D and calcitriol could block inflammatory pathways and treat pathogen infections in dry eye disease. Firstly, vitamin D may inhibit the mRNA expression of TLRs and can block NF-κB at the nucleus level. McDermott et al. [39] found that human corneal endothelial cells activate vitamin D and inhibit TLR-mediated inflammation, as well as upregulating IκB, which inhibits NF-κB expression [98]. Studies have also shown that the level of IκB phosphorylation was significantly increased under hyperosmolar conditions but reduced by 50% after treatment with calcitriol. Immunofluorescent staining showed that calcitriol significantly reduced NF-κB/p65 expression in the nucleus, thereby inhibiting the transcription of downstream factors at the nuclear level. The experiments by Zhang et al. [98] confirmed that a high osmotic pressure environment could significantly promote the secretion of proinflammatory cytokines such as IL6 and IL8 by iHCECs, while the mRNA levels of IL6, IL8, MIP1A and MIP1B were decreased by calcitriol from the mRNA levels, which was dose-dependent and did not affect the viability of cells.

To sum up, pathogens stimulate the activation of T-cell-mediated inflammatory pathways on the ocular surface, while, in hyperosmolar environments, inflammatory factors in corneal endothelial cells resulted in NF-κB being activated. Vitamin D may inhibit the mRNA expression of TLRs and can block the NF-κB at the nucleus level. The influence of vitamin D on the diversity of bacterial flora on the ocular surface should be explored, and further studies are needed on the possible treatment of bacteria-related dry eye with calcitriol, which may have a therapeutic effect [99].

##### Other Vitamin Deficiencies

Vitamin A also has vital effects on ocular surface diseases. There have been a lot of studies focusing on the correlation between vitamin A deficiency and dry eye. It is well accepted that vitamin A helps goblet cell proliferation and mucin expression, and the vitamin A deficiency shortens the TBUT, causing damage to lacrimal gland acinar cells [100]. A rabbit model of dry eye showed that the expression levels of NF-κB and NOS were prominently higher than in normal controls. We speculated that the occurrence of dry eye may be related to oxidative stress and NF-κB pathways [101]. Additionally, vitamin A affects the apoptosis of corneal epithelial cells. Zhang et al. showed that the TBUT and tear volume were significantly decreased in a BAC-induced mouse model, while, when a vitamin A microemulsion was used for 7 consecutive days, the TBUT and tear volume increased. Similarly, with the use of a vitamin A microemulsion, the mRNA of the proapoptotic gene Bax was downregulated and the antiapoptotic gene Bcl-2 was significantly upregulated, meaning vitamin A can suppress endothelial cell apoptosis by activating Bcl-2 and inhibiting the BAC [102]. Additionally, vitamin A protects the keratoconjunctival epithelium from squamatization, which can alleviate dry eye symptoms by promoting differentiation of the non-secretory epithelium into the secretory epithelium. The synthesis of glycoproteins in corneal epithelial cells was also promoted by vitamin A, suggesting that vitamin A exerts a prominent effect on corneal energy metabolism [102].

Moreover, Paolo F et al. [103] figured out that Vitamin A can upregulate the density of conjunctival goblet cells, promote mucin secretion, promote M16 and M4 secretion and improve the quality of tears. Moreover, vitamin A could reduce the keratinization of the ocular surface to relieve the inflammation of the conjunctiva.

We can see that vitamin A is strongly associated with dry eye through inflammatory pathways; however, no studies on vitamin A and bacteria have been found so far. We believe that vitamin A may promote the physiological barrier function and prevent bacteria from proliferating by inhibiting epithelial apoptosis. A lack of vitamin A results in damage to the lacrimal gland and secretory epithelium, which destroys the immune barrier. In addition, vitamin A inhibits the activation of the NF-κB pathway, and in turn may inhibit downstream inflammatory cytokines and the development of dry eye.

Vitamin C is another potential candidate in the development of dry eye. The tear film contains vitamin C, which is thought to help maintain the integrity of blood vessels and connective tissues, and to quell free radicals generated by the high metabolic activity [104]. Vitamin C likely acts as a non-enzymatic antioxidant and anti-inflammatory agent that appears to be effective in alleviating dry eye by downregulating TNF-α-induced ICAM-1 expression via the inhibition of NF-κB activation [105]. Additionally, it has been detected that the leucocyte ascorbic acid levels were significantly lower in patients with infectious corneal ulcers, which illustrates that vitamin C will protect the ocular surface from infection.

Currently, there is no evident relationship between vitamin C and bacteria on the ocular surface, although we suspect that dry eye induced by vitamin C deficiency results in a decline in immune function and a defect in the anatomical barrier of the ocular surface, causing the bacterial susceptibility to increase, as well as causing changes to the diversity of bacterial flora on the ocular surface; however, these hypotheses still require further proof.

Additionally, vitamin E is an unknown factor on the ocular surface. The mechanism remains unknown, although Costanza et al.’s study showed that the combination of crosslinked hyaluronic acid and coenzyme Q10 with vitamin E TPGS has the potential to protect the ocular surface from potential damage [106].

In recent years, the incidence of vitamin deficiency has gradually decreased due to the elimination of poverty in many regions and improvements in quality of life, although we should still pay attention to certain special groups, such as vegetarians and patients undergoing subtotal gastrectomy. Comprehensive assessments of the severity of vitamin deficiency and eye infection status should be conducted to avoid misdiagnosis.

## 2. Potential Treatment Targets

Based on the above, we suggest that the treatment of dry eye is not limited to relieving symptoms, but, more importantly, the underlying cause of the dry eye should be the basis for treatment. Regarding bacteria-related dry eye, several aspects should be considered before the choice of treatment methods is made.

In the first instance, an evaluation of the bacterial severity on the ocular surface should play a vital role, providing indications for the use of antibiotics, as was discussed above. The application of antibiotics is a double-edged sword, since it eliminates both commensal flora and foreign pathogens. For dry eye patients with mild bacterial flora changes, antibiotics are not recommended. Instead, artificial tear substitutes might be a good choice, which improve the conditions of hyperosmosis, contributing to the recovery of commensal bacteria. Additionally, the activation of the TLR stimulated by osmotic changes might also be reduced, which can lead to symptom relief. Meanwhile, antibiotics could be effective in patients with higher bacterial severity, which has been proven in patients with MGD in clinical practice [107].

Moreover, drugs targeting the downstream inflammation pathways of the bacteria in terms of the pathogenesis of dry eye are irreplaceable [108,109]. Additionally, the TLR and its downstream inflammation response have been identified as important factors in the pathogenesis of dry eye, and anti-inflammatory therapy could be quite effective. Currently, in clinical settings, steroid and non-steroidal anti-inflammatory drugs (NSAID) are applied in the treatment of intractable dry eye. Topical corticosteroids have been shown to improve the symptoms and clinical signs of moderate to severe dry eye, and, in the US, treatments targeting T cells for DED have been approved. Furthermore, 0.1% cationic ciclosporin reduced the inflammatory factor HLA-DR and reduced tear film osmolarity. Several newly developed anti-inflammatory drugs also seem promising for future clinical use. Tacrolimus can achieve anti-inflammatory effects by blocking the activity of T lymphocytes, while topical 0.03% tacrolimus eye drops for SS can significantly improve the fluorescein and Rose Bengal scores [110]. Secukinumab, a monoclonal antibody that binds IL-17A, successfully mitigates multiple aspects of the DED by inhibiting lymphangiogenesis and reducing B cell formation, while canakinumab, a monoclonal antibody targeting IL-1β, is now also being used [111]. ROS play a crucial role in the regulation of inflammatory responses. Polydatin, a monocrystalline compound, acts as an antioxidant, suppressing hyperosmolar-stress-induced cell cytotoxicity and inflammation via blockage of the NF-κB and NLRP3 pathways and ROS production. *Lactobacillus* species were detected in most patients with dry eye. Lactobacillus lysate (LBL) was used to inhibit the LPS-induced IL-1β and TNF-α release from macrophages [112].

Finally, other treatments might also be effective in relieving bacterial disturbances and related dry eye. Tear substitutes have been discussed above, which might improve the microenvironment of the ocular surface and reconstruct the commensal flora. Vitamin deficiency is directly related to structural damage, as well as to the activation of inflammatory factors. Proper vitamin supplementation might be crucial for dry eye treatment. Probiotics have been a research hotspot in recent years, and have been demonstrated to be effective in the treatment of colitis and inflammatory bowel disease; however, currently no studies have reported on the application of probiotics on the ocular surface, which requires further attention.

The potential treatment methods mentioned above still require exploration via cohort studies focused on bacterial changes with different treatment methods.

## 3. Conclusions and Prospects

To sum up, various factors are thought to be involved in the pathogenesis of dry eye, and a great deal of factors remain to be elucidated. Recently, disturbances of bacterial flora homeostasis on the ocular surface have been a research hot spot in dry eye progression. In this study, we systemically reviewed the current literature to summarize the bacterial differences between patients with dry eye and the general population, and identified several possible mechanisms.

For now, studies on the alteration of ocular surface bacteria and their role in the pathogenesis of dry eye are still at the preliminary stage [113]; further experiments focusing on the mechanisms of bacteria involved in dry eye progression are needed. On the one hand, it would be helpful for us to explore their pathogenesis by clarifying the changes in ocular flora and the relationships with the floral change in other organisms. A previous meta-analysis described a significantly higher *H. pylori* infection rate in Sicca syndrome patients with dry eyes, suggesting that there may be a correlation between intestinal diseases and ocular surface disease. More to the point, the relationship between the gut flora and ocular surface flora deserves further investigation [114]. Additionally, the gut–eye–lacrimal gland axis has been proposed as a possible mechanism. In mice, antibiotic-induced intestinal dysbiosis worsens dry eye by increasing the recruitment of effector T cells to the ocular surface [115]. The distinctive intestinal microbial community on the ocular surface can provide us with a better understanding of dry eye, as dry eye might be a manifestation of systemic diseases. On the other hand, the bacterial results from the recent studies could not comprehensively indicate the true composition of the commensal flora due to limitations in the detection methods used. Specimens collected from the ocular surface always have low bacterial concentrations and small sample amounts, which strictly limits the positive culture rates of the bacteria and is a key obstacle in studying ocular surface diseases. The development of techniques based on genetic analyses is a great improvement for the identification of bacterial strains and drug resistance. Recently, third-generation gene-sequencing technology has been applied in clinical practice. Nanopores, for example, have been used in bacterial identification in lower respiratory infection disease, with a sensitivity rate improvement of 96.6% and specificity-rate improvement of 41.7% as compared with culture. Moreover, this technique requires few samples (400 μL respiratory samples) and a shorter detection time (approximately 6 h), which might make it promising for use in ocular sample detection in the future.

rRNA gene high-throughput sequencing is more and more popular, and the wide application of this technology can accurately measure the number of types of bacteria, and Zhenhao Li et al. [24] also measured the content of bacteria on the eye surface of dry eye patients through RNA sequencing.

With the development of various testing methods, it is possible to treat dry eye with bacteria and control inflammation.

## Figures and Tables

**Figure 1 jcm-11-04019-f001:**
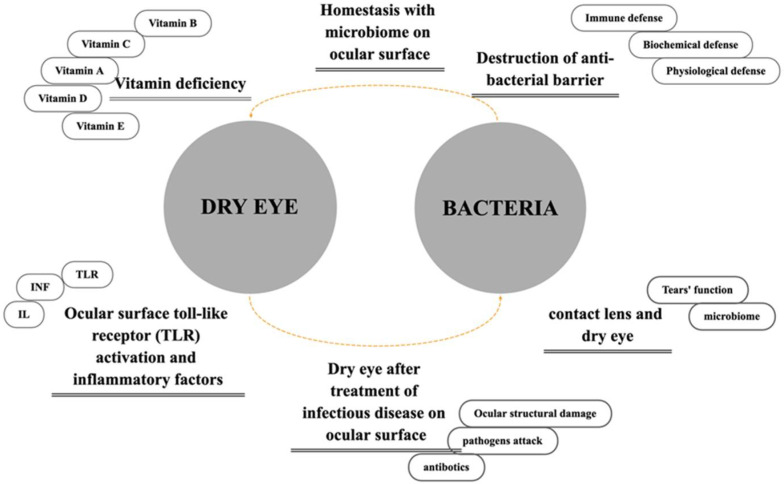
The mechanisms of dry eye disease.

**Figure 2 jcm-11-04019-f002:**
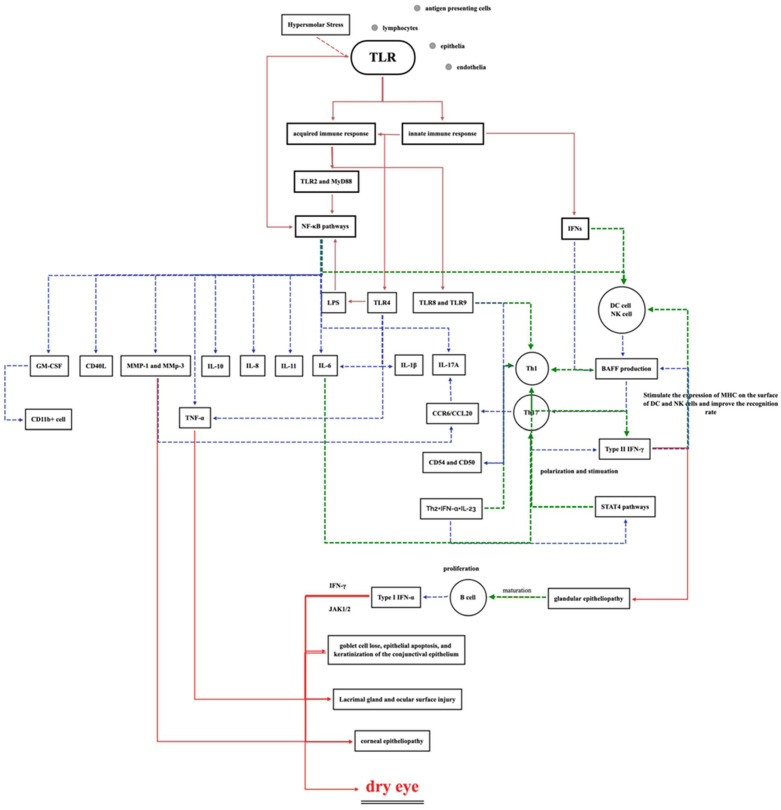
Possible molecular mechanisms involved in dry eye disease. TLR: toll-like receptor; MyD88: myeloid differential protein 88; IFN: interferon; NF-kB: nuclear factor kappa B; LPS: lipopolysaccharide; DC: dendritic cell; NK cell: natural killer cell; IL: interleukin; BAFF: B-cell-activating factor belonging to the TNF family; CCR: chemokine receptor; STAT4: signal transduction and transcription activator 4 antibody.

**Table 1 jcm-11-04019-t001:** Global comparison of the microbiomes of the general population and MGD population in recent years.

Authors	Country	*n*	Microbiome in General Population	Microbiome in Dry Eye Patients
Huang et al. [18] (2016)	China	31 healthy eyes	*Corynebacterium* (28.22%) *Pseudomonas* (26.75%) *Staphylococcus* (5.28%) *Acinetobacter* (4.74%) *Streptococus* (2.85%) *Millisia* (2.16%) *Anaerococcus* (1.86%) *Finegoldia* (1.68%) *Simosiella* (1.48%) *Veillonella* (1.00%)	
Doan et al. [19] (2016)	America	428 healthy eyes	*Coagulase-negative Staphyloccocus* (45.3%) *Propionibacterium* (33.9%) *Diphtheroids* (15.4%) *Streptococcus* (3.5%) *Coagulase-positive Staphylococcus* (2.1%) *Micrococcus* (2.1%) *Bacillus* (2.1%) *Lactobacillus* (0.2%) *Rothia* (0.2%) *Unidentified* *Gram-negative bacteria* (2.5%) *Neisseria* (0.9%) *Hemophilus* (0.5%) *Escherichia* (0.2%) *Enterobacter* (0.2%) *Moraxella* (0.2%)	
Watters et al. [29] (2016)	New Zealand	39	*S. aureus* (48.7%) *P. acnes* (25.6%) *Corynebact* sp. (1.3%) *Gm neg. rods inc. Pseudomonas* (5.1%)	*S. aureus* (30.3%) *P. acnes* (36.8%) *Corynebact* sp. (3.2%) *Streptococcus* sp. (3.5%)
Zhang et al. [27] (2017)	China	84 healthy eyes and 201 MGD eyes	*Staphylococcus epidermidis* (48.6%) *Corynebacterium macginleyi* (11.4%) *Staphylococcus lentus* (8.6%) *Staphylococcus hominis* (5.7%) *Staphylococcus lugdunensis* (5.7%)	*Staphylococcus epidermidis* (64.1%) *Staphylococcus lentus* (12.2%) *Staphylococcus aureus* (5.1%) *Corynebacterium macginleyi* (3.8%) *Staphylococcus homini* (3.2%) *Staphylococcus haemolyticus* (2.6%) *Corynebacterium tuberculostearicum* (1.9%)
Ozkan et al. [25] (2017)	Australia	43 healthy eyes	*Corynebacterium* (11.1%) *Acinetobacteria* (11.0%) *Pseudomonas* (10.4%) *Sphingomonas* (10.2%) *Streptococcus* (4.8%) *Massilia* (3.2%) *Rothia* (1.9%)	
Kara M et al. [22] (2017)	America	52 healthy eyes in children	*Staphylococcus* (56.5%) *Streptococcus* (16.9%) *Corynebacterium* (6.2%) *Moraxella* (8%) *Oceanospirillaceae* (7.32%) *Listeriaceae* (4.42 %) *Psychomonadaceae* (2.57%) *Leuconostocaceae* (2.07%)	
Jiang et al. [21] (2018)	China	58 healthy eyes and 82MGD eyes	*Staphylococcus (G+)* (13.6%) *S. epidermidis* (10.7%) *S. aureus* (1.4%) *S. hominis* (1.4%) *S. capitis* (0.7%) *Corynebacterium (G+)* (2.9%) *C. macginleyi* (2.9%) *Microbacteriaceae (G+)* (2.9%) *Microbacterium* (0.7%) *Micrococcaceae* (2.1%) *Moraxella osloensis (G−)* (2.1%)	*Staphylococcus (G+)* (47.9%) *S. epidermidis* (46.4%) *S. aureus* (2.9%) *S. hominis* (0.7%) *S. capitis* (1.4%) *S. warneri* (0.7%) *Corynebacterium (G+)* (4.3%) *C. macginleyi* (3.6%) *C. pseudodiphtheriticum* (0.7%) *Microbacteriaceae (G+)* (10.0%) *Microbacterium* (2.1%) *Micrococcaceae* (5.0%)
Li et al. [24] (2019)	China	54 healthy eyes and 35 dry eyes	*Proteobacteria* (51.70%) *Firmicutes* (16.86%) *Bacteroidetes* (13.60%) *Actinobacteria* (6.12%) *Cyanobacteria* (1.72%) *Acidobacteria* (1.66%) *Chloroflexi* (1.54%) *Planctomycetes* (1.43%) *Epsilonbacteraeota* (1.25%) *Verrucomicrobia* (1.06%)	*Proteobacteria* (47.62%) *Firmicutes* (17.20%) *Bacteroidetes* (16.54%) *Actinobacteria* (6.24%) *Cyanobacteria* (2.01%) *Acidobacteria* (1.69%) *Chloroflexi* (1.58%) *Planctomycetes* (1.40%) *Epsilonbacteraeota* (1.00%) *Verrucomicrobia* (0.95%)
Dong et al. [23] (2019)	China	42 healthy eyes and 47 MGD eyes	*Corynebacterium* (46.43%) *Staphylococcus* (7.88%) *Sphingomonas* (0.79%) *Snodgrassella* (3.60%) *Propionibacterium* (5.44%) *Streptococcus* (3.89%)	*Staphylococcus* (20.71%) *Corynebacterium* (20.22%) *Propionibacterium* (9.29%) *Sphingomonas* (5.73%) *Snodgrassella* (4.17%) *Streptococcus* (2.80%)

## Data Availability

Not applicable.

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
