# Peer review of "Bacteria and Dry Eye: A Narrative Review"

_jcm, 2022, doi:10.3390/jcm11144019_

Round 1
Reviewer 1 Report
The author tried to link the Ocular surface microbiota and the potential link with dry eye disease, however, so solid evidence was provided. Second, the author focuses on the ocular microbiota which can be cultured, how about studies that may have analyzed data with non-cultured techniques such as pyrosequencing? those studies may have provided more in-depth insight into the potential role of ocular microbiota and dry eye disease. Similarly, throughout the paper, the author failed to provide solid evidence for each argument. Here is another example, the author is talking about vitamin A and what role it plays on the ocular surface which is a general statement. Vitamin A has active components 9-cis retinoic acid and All-trans retinoic acid, the 9Cis binds to the RXR and AT-RA binds to the RAR receptors and they may play different roles. My suggestion is to revisit the manuscript and try to find papers that provides solid evidence, not speculation.
Reviewer 2 Report
In the current review, Wang et al. discussed the role of bacteria in the disease pathology of dry eye. The authors discussed the mechanisms of dry eye. The following are my major queries.
1) Dry eye is a multifactorial disease and is not only associated with changes in bacterial flora. There can be various hormonal or autoimmune diseases, environment, medication, aging, sex, etc. associated with dry eye. Is the viral infection also associated with dry eye? In the introduction, authors should mention briefly the various factors associated with the dry eye and afterward can focus on the role of ocular bacteria in the dry eye.
2) Authors discussed various clinical studies of ocular flora, in various ethnicity is there any single bacterial species associated with the dry eye?
3) Are there any studies of animal models of bacterial infection-induced dry eye models?
4) Which cells play a key role in dry eye disease pathology? Authors should describe the brief role of residential and infiltrating cells in dry eyes.
5) Authors should add the future directions in the current review.
Round 2
Reviewer 1 Report
As I mentioned previously, the manuscript is very descriptive and based on assumptions, the claims are not supported by scientific data e.g. in the following paragraph the author gives just only one reference, and when I read the whole paper it was about the gut and gut microbiota not about the ocular surface "
“Regarding the role of vitamin B in bacterial flora on the ocular surface in the
development of dry eye, several studies have provided insights in this field and might
outline implications for clinical practice. First of all, riboflavin, also known as vitamin B2,
has the potential to maintain the stability of physiological barriers consisting of microvilli
and microplicae in the superficial epithelium of the conjunctiva, leading to a decreased
number of goblet cells. This deficiency of vitamin B12 decreases the number of CD8+ T
and NK cells, which damage the immune barriers and weaken the immune response,
causing pathogen invasion. Thirdly, the mucosal-associated invariant T cells can be
activated to promote the production of IFN-γ and IL-17 by the vitamin B2 metabolite 6-
hydroxymethyl-8-D-ribityllumazine, which was produced by pathogens that bind to the
major histocompatibility complex (MHC)-related protein (MR1) on antigen-presenting
cells[98]”.
I would strongly suggest shortening the manuscript just to the ocular surface microbiota part, and also changing the title as the literature is not supporting the current title "Bacterial pathogenic dry eye and possible mechanisms".
Author Response
As I mentioned previously, the manuscript is very descriptive and based on assumptions, the claims are not supported by scientific data e.g. in the following paragraph the author gives just only one reference, and when I read the whole paper it was about the gut and gut microbiota not about the ocular surface "
“Regarding the role of vitamin B in bacterial flora on the ocular surface in the
development of dry eye, several studies have provided insights in this field and might
outline implications for clinical practice. First of all, riboflavin, also known as vitamin B2,
has the potential to maintain the stability of physiological barriers consisting of microvilli
and microplicae in the superficial epithelium of the conjunctiva, leading to a decreased
number of goblet cells. This deficiency of vitamin B12 decreases the number of CD8+ T
and NK cells, which damage the immune barriers and weaken the immune response,
causing pathogen invasion. Thirdly, the mucosal-associated invariant T cells can be
activated to promote the production of IFN-γ and IL-17 by the vitamin B2 metabolite 6-
hydroxymethyl-8-D-ribityllumazine, which was produced by pathogens that bind to the
major histocompatibility complex (MHC)-related protein (MR1) on antigen-presenting
cells [98]”.
I would strongly suggest shortening the manuscript just to the ocular surface microbiota part, and also changing the title as the literature is not supporting the current title "Bacterial pathogenic dry eye and possible mechanisms".
Dear Reviewer,
Thank you for the advice, I have reviewed and corrected them. I'm sorry that there are so many questions. If you have any other questions, please let me know. Thank you very much.
This manuscript is a resubmission of an earlier submission. The following is a list of the peer review reports and author responses from that submission.